# Activated LoRA: Fine-tuned LLMs for Intrinsics

**Kristjan Greenewald, Luis Lastras, Thomas Parnell, Vraj Shah, Lucian Popa, Giulio Zizzo,**
**Chulaka Gunasekara, Ambrish Rawat, David Cox**
IBM Research

## Abstract

Low-Rank Adaptation (LoRA) has emerged as a highly efficient framework for finetuning the weights of large foundation models, and has become the go-to method for data-driven customization of LLMs. Despite the promise of highly customized behaviors and capabilities, switching between relevant LoRAs in a multiturn setting is inefficient, as the key-value (KV) cache of the entire turn history must be recomputed with the LoRA weights before generation can begin. To address this problem, we propose Activated LoRA (aLoRA), an adapter architecture which modifies the LoRA framework to only adapt weights for the tokens in the sequence *after* the aLoRA is invoked. This change crucially allows aLoRA to accept the base model's KV cache of the input string, meaning that aLoRA can be instantly activated whenever needed in a chain without recomputing the prior keys and values. This enables building what we call *intrinsics*, i.e. specialized models invoked to perform well-defined operations on portions of an input chain or conversation that otherwise uses the base model by default. We train a set of aLoRA-based intrinsics models, demonstrating competitive accuracy with standard LoRA while significantly improving inference efficiency. We contributed our Activated LoRA implementation to the Huggingface PEFT library.[1]

## 1 Introduction

The rapid adoption of large language models (LLMs) has catalyzed significant advancements in natural language processing tasks, from text generation to knowledge extraction. However, adapting these models to specific tasks or domains often demands finetuning their immense parameter space, a process that is computationally expensive and difficult to scale. Low-Rank Adaptation (LoRA)[13] has addressed these challenges by introducing a parameter-efficient method for fine-tuning [12], enabling highly customized model behavior without the need to retrain or modify the entire model. By optimizing a small subset of low-rank matrices, LoRA has emerged as a widely-used lightweight and effective alternative for task-specific customization [25], particularly for large foundation models such as LLMs, and popular LoRA finetuning services are offered by both corporate and open-source LLM providers [26, 36, 28]. While large models perhaps have less need for finetuning, small models continue to see strong benefits from LoRA adapters [40, 5].

Motivated by this, significant work has been done to enable highly efficient serving of and inference with LoRAs, e.g. [34, 2], with vLLM [16] incorporating and further optimizing many of these algorithms in its popular, state-of-the-art inference engine. Yet, while LoRA excels in static or single-task scenarios, in modern applications a wide mix of skills is typically needed, e.g. in multiturn interactions and agentic applications [41, 44]. Crucially, LoRAs often must be carefully tuned to avoid forgetting/degradation of performance on the wide variety of tasks the base model already performs well on. In applications where dynamic switching between multiple specialized skills would be advantageous (e.g. agentic or reasoning pipelines), this strategy breaks down. Additionally, multi-task LoRA training is significantly more difficult than training a LoRA for a single task, and

---

[1] https://github.com/huggingface/peft

this approach would be inherently non-modular, requiring retraining to incorporate further abilities later on.

In an ideal world, we would like to be able to seamlessly switch—within the same chat/agentic pipeline—between arbitrary sets of pretrained LoRA configurations for specialized tasks, while keeping the base model as-is for most interactions in the sequence. Unfortunately, the LoRA architecture is not well-suited to this regime, creating major inference inefficiencies. Specifically, for each switch between LoRA adapters, it becomes necessary to recompute the representation of all context tokens prior to generation with the LoRA. In the case of the popular attention mechanism introduced by [37], such a representation takes the form of the key-value (KV) cache of such prior tokens[2], but more generally, it is a problem that reoccurs even in different architectures; for example, in a state-space model, the state representation of the tokens prior to generation would need to be recomputed if the matrices $A$ and $B$ are updated by a low rank correction. This recalculation introduces significant latency, GPU memory, and computational overhead that all scale with the length of the context that must be prefilled, especially as LLMs increasingly work with documents, files, or reasoning/agentic chains in the many (sometimes hundreds of) thousands of tokens [24, 21, 22] (even millions for software engineering [14]). This limits LoRA's usability in scenarios where rapid transitions between specialized behaviors are essential.

In this work, we address this shortcoming, presenting our Activated LoRA method, which *activates* adapted weights only on tokens corresponding to a short intrinsic instruction and subsequent generation, leaving the weights for other context tokens unchanged. We elaborate on the setting we envision here. Within the discipline of software engineering, *intrinsics* are generally useful functions that are built into a programming language whose implementation can be optimized by a compiler. We define an LLM intrinsic to be a capability that can be invoked through a well defined API that is reasonably stable and independent across model generations and families of how the intrinsic is implemented. Performance metrics such as accuracy, latency, and throughput may vary significantly across such implementations.

The concept of activated LoRA takes an opinionated view on how such differentiated performance could be attained. As inspiration, we note that instructions in LLMs can appear in many places in a prompt; Figure 1 illustrates an example of two such places: an "early prompting" case where the instruction precedes the content, or a "late prompting" case where the content precedes the instruction. In the latter, the instruction does not have to be revealed ahead of time, making the representation of the content (KV cache) potentially reusable for or from other tasks. We note that these paradigms can take on many shapes; for example, prefix tuning methods such as [18] explicitly use an early prompting framework, tuning this early prompt. Recently, a set of real-world tasks (and trained adapters) conforming to the late-prompting intrinsics regime was presented in [5], in the context of building performant and robust RAG [17] pipelines. We believe that a vast array of intrinsics-style finetuning tasks can be created with applications throughout the agentic, chat, and reasoning spaces.

A LoRA adapter, like the prefix tuning example above, shares the downsides of the early prompting concept; the LoRA-adapted LLM's internal representation of the content (KV cache) is fit-for-purpose, specific to the LoRA adapter and not reusable by or from the base model or other adapters. Note that while late-prompt tuning approaches do exist and may be viable in some cases, they tend to underperform prefix tuning (early-prompt tuning), which in turn significantly underperforms LoRA-based training [10]. As a result, a trainable late-prompting style adapter is needed to close this gap.

We therefore introduce Activated LoRA (aLoRA), a novel extension of the LoRA framework designed to only adapt the model's weights for tokens encountered *after* activation, allowing base model KV cache for prior context to be reused (Figure 2). By decoupling the adaptation process from the need to recompute the input's representation, aLoRA facilitates instantaneous switching between specialized models (intrinsics) while maintaining seamless interaction with the base model. This innovation not only reduces computational costs, but also unlocks new possibilities for deploying highly modular, task-specific behaviors within complex workflows.

---

[2]In this work, we use the term "(KV) cache" to denote the set of saved keys and values for prior tokens in the LLM context, noting that our approach is not limited to attention types that use keys and values. We use "(KV) cache reuse" to mean (re)use of stored keys and values from an LLM call (e.g. base model generation) in a subsequent LLM call (e.g. the aLoRA generation) whose context shares a prefix with the original context and/or generation.

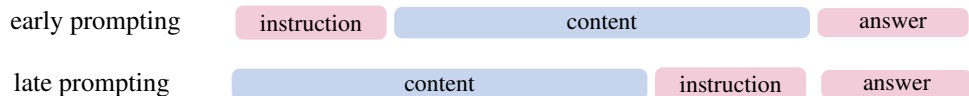

Figure 1: Late vs. early prompting framework for intrinsics. The aLoRA adapter architecture is designed to preserve the cache-reuse benefits of *late prompting* by adapting weights only on the red tokens, allowing it to reuse the base model cache for the blue input tokens.

In our experiments, we demonstrate significant speedups for aLoRA vs LoRA on the state-of-the-art inference engine vLLM [45], and show that aLoRA adapters do not lose accuracy versus LoRA on a collection of benchmark finetuning tasks and a collection of real-world "intrinsics" tasks from [5].

**Relationship to other LoRA variants.** QLoRA [7] proposes quantizing the weights in the base model while keeping the adapter higher precision in training and inference, which dramatically reduces overall memory costs. This can be directly applied to Activated LoRA as well[3], with KV cache reuse still possible if the base model inference also uses the same or sufficiently similar quantization. Experiments with this are beyond the scope of the present work. DoRA [23] adapters are similar to LoRA with a different decomposed low rank weight matrix, with magnitude and direction of the adaptation parameterized separately. In principle, extension to DoRA-style adapters ("Activated DoRA") is immediate along the same principles as aLoRA. That said, inference with DoRA is less efficient than LoRA, so it is typically recommended [25] that DoRA adapters be merged into the base model weights — a procedure that is not possible for our "activated" approach due to the selective application of the adapter weights.

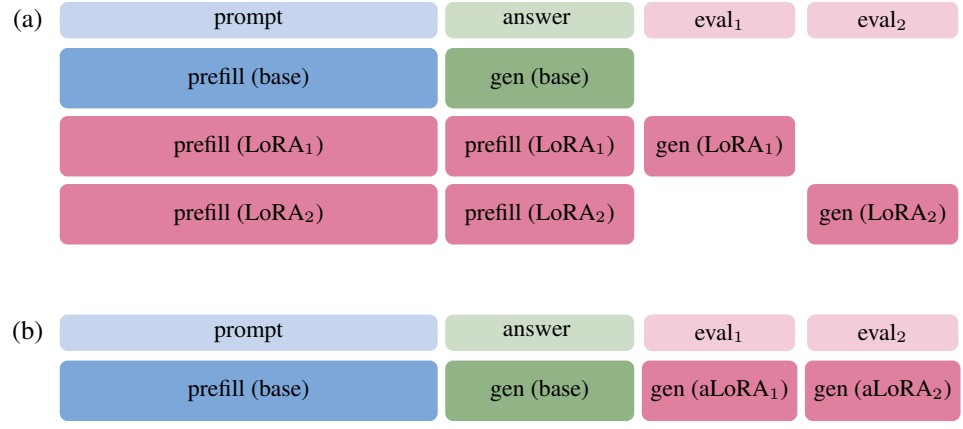

Figure 2: Computation and memory pattern of (a) LoRA vs. (b) aLoRA used as evaluators of an answer given by a base model. (1) `prompt` is input to the base model, which generates `answer`, (2) `prompt + answer` is input to both intrinsics in parallel, which generate `eval_1` and `eval_2` respectively. Narrow rectangles denote tokens and wide rectangles denote the KV cache.

## 1.1 Background

Modern decoder-only causal LLMs are transformers containing a sequence of decoder layers, each of which typically contain MLP layers that operate on each token's representation independently, and a causal attention mechanism that allows tokens to attend to representations of prior tokens in the same layer (see Figure 9 in the appendix). We review the ubiquitous softmax attention mechanism introduced by [37] and its connection to the KV cache, as well as LoRA adapters.[4] Further, note that LoRA and aLoRA adapters can be applied to MLP blocks as well, but since it is typical practice to not adapt the MLP blocks we focus the presentation on the attention blocks only.

---

[3]Implementation already supported in the Huggingface PEFT library.

[4]Extension of this to other forms of attention is straightforward, but we leave this treatment to future work.

**Attention:** Recall that the softmax attention mechanism in each attention layer takes the form

$$\text{Attention}(Q, K, V) = \text{softmax}\left(\frac{QK^\top}{\sqrt{d_k}}\right) V, \tag{1}$$

where $d_k$ is the dimension and $Q, K, V$ are concatenated queries, keys, and values for the tokens:

$$Q = XW^Q, \quad K = XW^K, \quad V = XW^V \tag{2}$$

where $W^Q$, $W^K$, and $W^V$ are weight matrices.

**LLM inference and the KV cache:** LLMs, using a causal attention mask [29], generate tokens autoregressively one at a time. Suppose we have already generated tokens $1, \ldots, t-1$ with corresponding hidden-state embeddings $x_1, \ldots, x_{t-1}$. Whenever these tokens were processed (either during prefill or generation) their keys and values were already computed:

$$K_{1:t-1} = [K_1; \ldots; K_{t-1}], V_{1:t-1} = [V_1; \ldots; V_{t-1}], \quad K_i = x_i W^K, V_i = x_i W^V.$$

Due to causal attention, tokens can only attend to prior tokens, and hence are not affected by subsequent generations and can be stored for use by future tokens. At generation step $t$, we therefore only need to compute the new query, key, and value for the latest token

$$Q_t = x_t W^Q, \quad K_t = x_t W^K, \quad V_t = x_t W^V, \tag{3}$$

and then form the full cached matrices by appending $K_{1:t} = [K_{1:t-1}; K_t], V_{1:t} = [V_{1:t-1}; V_t]$. Finally, the output hidden state for token $t$ is ($d_k$ is hidden dimension)

$$h_t = \text{Attention}(Q_t, K_{1:t}, V_{1:t}) = \text{softmax}\left(\frac{Q_t K_{1:t}^\top}{\sqrt{d_k}}\right) V_{1:t}. \tag{4}$$

Because the entire matrix of past keys $K_{1:t-1}$ and values $V_{1:t-1}$ is precomputed and stored in the "KV cache," at each new step $t$ we only pay the cost of computing one new row for each of $Q$, $K$, and $V$, and performing a single-row softmax and weighted sum over the remaining cached $K$ and $V$. As shown in [37, 29], this reduces what would have been an $\mathcal{O}(t^2 d_k)$ full-matrix multiply down to only $\mathcal{O}(t\, d_k)$ operations at each step, yielding massive speedups.

**LoRA:** LoRA adapts the attention weights $W^Q$, $W^K$, and $W^V$ by replacing them with $W^Q + \Delta^Q$, $W^K + \Delta^K$, and $W^V + \Delta^V$, where $\Delta^Q, \Delta^K, \Delta^V$ are rank $r$ matrices. This yields

$$Q = X(W^Q + \Delta^Q), \quad K = X(W^K + \Delta^K), \quad V = X(W^V + \Delta^V). \tag{5}$$

This lowers the number of parameters, making finetuning significantly more efficient [13]. If the LoRA is active for the entire chain, then the KV cache-based inference applies and generation is efficient. If, on the other hand, any part of the input was generated or prefilled by the base model or another LoRA, then $K$ and $V$ for the LoRA are different than the corresponding $K$ and $V$ for the base model, hence any existing base model KV cache cannot be used by the LoRA, meaning the KV cache for the entire (potentially very long) input must be recomputed. To clarify, this involves passing the input through all layers of the transformer in sequence–not just the adapted attention layers–since modifications to attention blocks modify the inputs to all downstream layers and tokens.

## 2 Activated LoRA

Just as in LoRA, our aLoRA architecture adapts the attention weights $W^Q$, $W^K$, and $W^V$ by replacing them with $W^Q + \Delta^Q$, $W^K + \Delta^K$, and $W^V + \Delta^V$, where $\Delta^Q, \Delta^K, \Delta^V$ are rank $r$ matrices. The difference lies in how these adapted weights are used. We assume that the default generation model for the chat is the base model, and that intrinsics only operate on these base model generations. As a result, we can assume that the base model has precomputed a KV cache for the input context (the blue region in Figure 1).

The aLoRA architecture is designed to **match the base model keys and values on context tokens**, allowing the adapter to reuse base model KV cache for those tokens, or, vice versa, the base model to reuse KV cache that the aLoRA adapter creates for its context inputs. Specifically, in the attention mechanism (1), aLoRA only adapts the $Q, K, V$ matrices for tokens occurring after the start of the

invocation sequence. Let the token index for adapter activation be $t_{\mathrm{invoke}}$, and let the current token have index $t \geq t_{\mathrm{invoke}}$. Instead of (5) we then have

$$Q = \begin{bmatrix} X_{1:t_{\mathrm{invoke}}-1}W^Q \\ X_{t_{\mathrm{invoke}}:t}(W^Q+\Delta^Q) \end{bmatrix}, K = \begin{bmatrix} X_{1:t_{\mathrm{invoke}}-1}W^K \\ X_{t_{\mathrm{invoke}}:t}(W^K+\Delta^K) \end{bmatrix}, V = \begin{bmatrix} X_{1:t_{\mathrm{invoke}}-1}W^V \\ X_{t_{\mathrm{invoke}}:t}(W^V+\Delta^V) \end{bmatrix} \quad (6)$$

where $X_{1:t_{\mathrm{invoke}}-1}$ and $X_{t_{\mathrm{invoke}}:t}$ are the portions of $X$ coming before and after the aLoRA model is invoked. If $X_{1:t_{\mathrm{invoke}}-1}$ is associated with tokens generated or prefilled by the base model, then $X_{1:t_{\mathrm{invoke}}-1}W^K$ and $X_{1:t_{\mathrm{invoke}}-1}W^V$ are already in the KV cache and do not need to be recomputed. Similarly, any tokens that have been prefilled or generated by the aLoRA model have keys and values processed by the adapted weights, so $X_{t_{\mathrm{invoke}}:t}(W^K + \Delta^K)$ and $X_{t_{\mathrm{invoke}}:t}(W^V + \Delta^V)$ are also available (except for the current token being generated). As a result, the aLoRA architecture can seamlessly reuse the existing base model KV cache as well as continue to maintain its own KV cache as it generates. Note that adaptations for the MLP blocks that preserve this KV cache property can be done via directly corresponding equations as for the $Q, K, V$ blocks above.

We can formalize this cache reuse claim as follows (proved in the appendix):

**Proposition 1** (KV equivalence and aLoRA inference). *For the causal decoder-only transformers we consider, the keys and values (actually, all internal states) prior to $t_{\mathrm{invoke}}$ are identical for the base model and any aLoRA adapter model using (6). Specifically, $K_{1:t_{\mathrm{invoke}}-1}^{\mathrm{base}} = K_{1:t_{\mathrm{invoke}}-1}^{\mathrm{adapter}}$, and $V_{1:t_{\mathrm{invoke}}-1}^{\mathrm{base}} = V_{1:t_{\mathrm{invoke}}-1}^{\mathrm{adapter}}$. Inference with the aLoRA adapted model can be done causally one token at a time (by simply increasing $t$ in (6) iteratively) with KV cache reuse.*

Furthermore, we can quantify the computation and memory savings:

**Proposition 2** (aLoRA vs. LoRA inference costs). *Consider invoking an adapter with $T_{\mathrm{cache}}$ tokens of input for which a base model KV cache exists and $T_{\mathrm{new}} \ll T_{\mathrm{cache}}$ input tokens without cache.[5] The first token generated by the aLoRA adapter requires $O(T_{\mathrm{cache}}T_{\mathrm{new}})$ operations, while LoRA requires $O((T_{\mathrm{cache}} + T_{\mathrm{new}})^2)$. Furthermore, aLoRA must maintain only $O(T_{\mathrm{new}})$ additional KV cache memory, while LoRA requires additional $O(T_{\mathrm{cache}} + T_{\mathrm{new}})$. If $N$ distinct adapters share the first $T_{\mathrm{cache}}$ tokens as input, aLoRA costs become $O(NT_{\mathrm{cache}}T_{\mathrm{new}})$ and $O(NT_{\mathrm{new}})$, while LoRA has $O(N(T_{\mathrm{cache}} + T_{\mathrm{new}})^2)$ and $O(N(T_{\mathrm{cache}} + T_{\mathrm{new}}))$.*

As the context length and number of concurrent adapters rise, the computation and memory advantages of aLoRA scale linearly. Note that while methods exist for KV cache compression that can reduce overall memory footprint, e.g. [19], and various methods exist increasing the practical scalability of attention, KV cache reuse is still a major factor in practice as context length increases.

**Cache reuse patterns:** Importantly, while we typically emphasize the aLoRA adapter reusing cache from the base model, the fact that input cache is interchangeable between the base model and *every* aLoRA adapter presents multiple opportunities for reuse. Consider three regimes:

1. The adapter reuses cache from the base model, e.g. checking a base model generation.
2. The base model reuses cache created when the aLoRA adapter was invoked, e.g. when an adapter checks an input prior to sending to the base model for full generation.
3. Multiple adapters reuse *the same cache*, e.g. when checking base model outputs on multiple axes. This regime creates the most dramatic advantage for aLoRA over LoRA.

In the results, we explore concrete intrinsics which follow these patterns. Note that any KV cache created by aLoRA adapter for their own generations are *not* reusable by other adapters or the base model, as these tokens come *after* the adapter weights are turned on. If these strings needs to be input into other adapters or the base model, they should be prefilled again. This is a fairly minor concern, however, as intrinsics generations are often short, and per token, prefill is much faster than generation.

**Invocation:** We found it useful to demarcate the activation of the aLoRA adapter via a short *invocation token sequence*. Advantages are elaborated on in Appendix A and include allowing more space for the adapted weights to process the input. Typically, this sequence will be appended to the prompt prior to generation, just as a "generation prompt" is typically appended to prompts for instruct-tuned model (e.g. `<|assistant|>` or some other string). While the aLoRA will often be invoked programmatically, this design in principle allows for the base model (or other intrinsics) to call the aLoRA model themselves (we do not explore this behavior in the current work).

---

[5]We presume that $T_{\mathrm{new}}$ is the invocation sequence and following.

## 2.1 Training

The activated LoRA framework is explicitly designed for tuning the LLM to modify its output conditioned on an input. This regime matches (but is not limited to) instruction finetuning tasks, where the context tokens are excluded from the loss while finetuning the adapter to produce a specified output sequence (supervised finetuning, or SFT). aLoRA can in principle also be used in RL training pipelines, though we do not explore these in the current work.

Our implementation of aLoRA is in the supplementary material. It seamlessly supports standard Huggingface training libraries [43] as well as inference/generation methods (with or without using available base model KV cache) if needed (e.g. for testing or proofs of concept when the efficiencies of vLLM [45] are not needed). Later in this section, we will also show inference experiments using our modification of the more efficient (SOTA) inference package vLLM to support aLoRA.

For aLoRA SFT, training data is specified as a set of (possibly multiturn) inputs and completions, typically with the base model's chat template applied. An invocation token sequence for the aLoRA model is optionally specified as described in the previous section. This invocation sequence is appended to the input sequence, and the model is finetuned to produce the output given the input. To train the aLoRA adapter, a base model is specified, and following standard LoRA practice, we apply low-rank adapters to any (or all) of the query, key, and value blocks in the attention layers.[6] In training, the aLoRA is aware of which tokens occur before the invocation sequence, and does not adapt the weights for those tokens (as in (6)).

**Adapter Model Capacity and Increased Rank:** Empirically, we observe that aLoRA adapters sometimes require higher rank than LoRA adapters to get good performance (no experiments in this paper use rank higher than $r = 32$). We offer the following intuition for this. LoRA adapters are free to adapt the weights for the keys and values for tokens prior to activation, so they are able to "compress" information needed for generation into the low-rank signal captured by the adaptation. This signal can then be more easily "picked up" by the adapted query weights for generated tokens. See Appendix D for further exploration. This also provides the key motivation for starting the adapted weights at the intrinsic instruction tokens, rather than waiting to activate the weights only at generation time—this choice boosts model capacity.

We now compare aLoRA to standard LoRA both in terms of inference compute costs as well as generation quality and accuracy. Recall that our goal is to achieve significant gains in inference cost, while preserving (not losing) accuracy relative to what is achievable with LoRA.

## 3 Inference timing results on vLLM

In Figure 2 we illustrate how computation and memory differ in LoRAs versus aLoRA in a simple agent pattern. A prompt (blue) is passed to a model which then prefills the corresponding KV cache, and then generates an answer (green). The task is now to evaluate the answer; in this example, two different hypothetical evaluators are used. An example of such evaluations may be to determine if the answer is faithful to the content provided, or how certain the model is of the answer, given the content. If these evaluators are implemented using LoRAs (Figure 2a), then to generate the evaluations, new KV cache computations need to be performed, independently for each evaluator. In the case of aLoRAs (Figure 2b), the KV cache of the underlying model is reused, resulting in significant savings.

To prove this experimentally, we modified vLLM [45] to be able to perform inference on aLoRAs with base model prefix cache reuse, and compared LoRA and aLoRA inference on 7 small ($\leq$ 14B) LLMs in the setting where the initial base model prompt length is varied, the base model generated answer is comprised of 256 tokens, and each adapter evaluation is given by 16 tokens.[7] Results are in Figure 3, showing timing for both 1 evaluator and 5 evaluators.[8] In this experiment, we set the LoRA to have rank 8 and the aLoRA to be rank 32, illustrating that any need for rank increase with aLoRA has negligible inference time effect since the adapter parameters are still typically much less than 1% of the base model parameters.

---

[6]Adapting MLP layers is possible but less common in LoRA SFT.

[7]Code for our vLLM implementation can be found at `https://github.com/tdoublep/vllm/tree/alora`.

[8]While 5 evaluators may initially seem a lot, this is a stand-in for more complex pipelines where the input, retrieved documents, and generation are all being checked, as in [5].

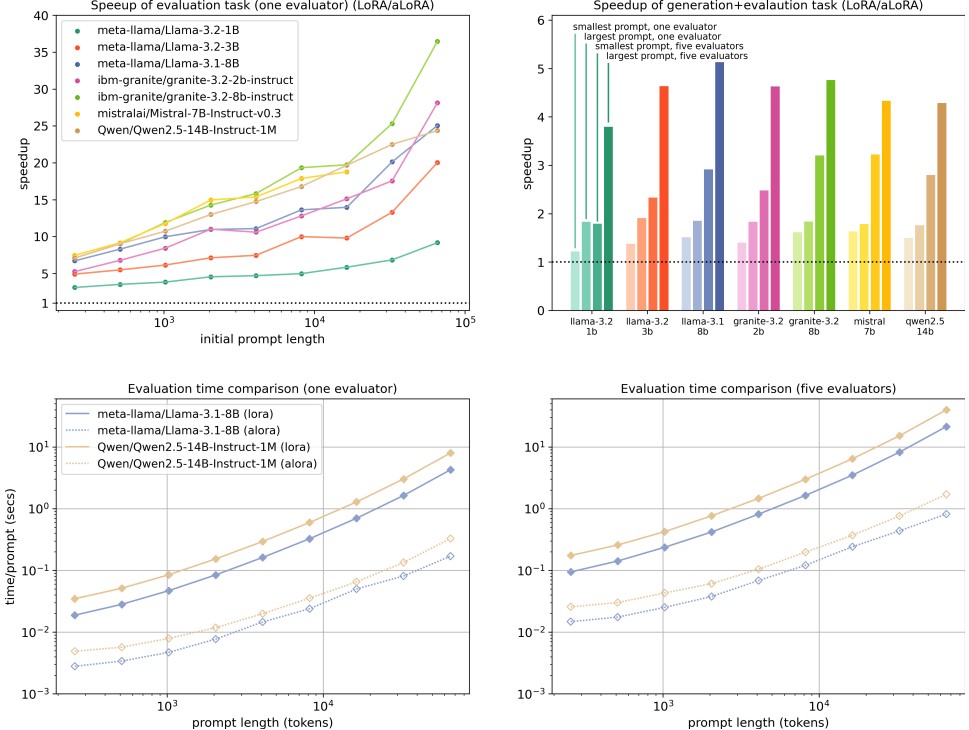

Figure 3: Comparison of aLoRA and LoRA when used as evaluators in a simple agentic pattern. **Top left**: Multiplicative speedup of an aLoRA evaluator vs LoRA, showing up to 35x improvement depending on base model and prompt length. **Top right**: Multiplicative speedup for the end-to-end pipeline including the base model generation (256 tokens) and 1 or 5 parallel eval (adapter) generations (16 tokens each). Despite the large fixed cost of the base model call, end-to-end aLoRA speedups are still significant, highlighting LoRA inefficiency. **Bottom row**: Log-log plots for the wall clock evaluation, showing that even for small models, the delay for LoRA becomes significant in absolute terms as the prompt and number of evaluations in the agentic pipeline scale.

The speedup factor of aLoRA increases as prompt length increases, adapter generation time improving over 2 to 7× even for 250-token prompts, and over 20× for most models on the longest prompts. Despite most of the compute advantages coming from the prefill savings (time to 1st token of the adapter), we still see highly meaningful speedups in the end-to-end pipeline which also includes the large fixed cost of generating 256 tokens with the base model. Finally, in the last row, despite the adapters only generating 16 tokens, we see that the LoRA delays are noticeable in wall clock time. Note these wall clock delays will be further magnified for more complex agentic or reasoning pipelines where many rounds of input and generation happen before the overall pipeline completes.

## 4   Finetuning Accuracy Results

Having demonstrated significant inference time speedups for aLoRA, it remains to ascertain whether we lose generation accuracy or quality. We first train LoRA and aLoRA adapters on 4 LLMs on a set of benchmark SFT tasks, and then consider a set of more challenging "intrinsics"-type tasks for which well-engineered, very recent LoRA adapters already exist against which we can compare. Our code implementing Activated LoRA has been contributed to the Huggingface PEFT library `https://github.com/huggingface/peft` [25], which we also use for LoRA.[9]

| Task | Llama 3.2 1B | | Llama 3.2 3B | | Llama 3.1 8B | | Mistral 7B | |
|---|---|---|---|---|---|---|---|---|
| | LoRA | aLoRA | LoRA | aLoRA | LoRA | aLoRA | LoRA | aLoRA |
| Bengali Hate Speech Classification | 79.30% | 81.94% | 86.34% | 89.43% | 70.04% | 85.02% | 72.25% | 85.46% |
| WIQA: Effect Classification | 68.92% | 71.38% | 76.15% | 76.00% | 74.92% | 78.00% | 61.08% | 79.08% |
| MMLU Conceptual Physics MCQA | 33.33% | 38.89% | 72.20% | 66.67% | 55.56% | 55.56% | 55.56% | 55.56% |
| MMLU College Computer Science MCQA | 66.67% | 58.33% | 66.67% | 75.00% | 66.67% | 58.33% | 75.00% | 75.00% |
| SocialIQA Question Generation | 86.00% | 88.77% | 89.85% | 90.15% | 52.00% | 88.92% | 97.23% | 90.92% |
| Hindi Sentence Perturbation | 69.60% | 74.69% | 98.30% | 63.89% | 86.11% | 35.19% | 99.23% | 96.30% |
| SuperGLUE Question Generation | 98.42% | 95.79% | 95.26% | 96.84% | 98.95% | 92.11% | 99.47% | 92.11% |

Figure 4: LoRA vs. aLoRA accuracy (%) on each task across base models after hyperparameter grid search guided by the validation set. While individual task performance is noisy due to the size of the datasets etc., there is no consistent accuracy loss from using aLoRA over LoRA.

## 4.1 Benchmark SFT tasks

In [2], a collection of 1000 instruction SFT tasks were curated and LoRA adapters were trained for each. These tasks were drawn from the Super-Natural Instructions [42] benchmark collection of datasets, which in turn drew from sources such as MMLU [11] etc.

From this list, we selected 7 tasks at random, roughly split between multiple-choice and freeform outputs. We excluded from consideration any tasks with (a) too small datasets, (b) overly open-ended instructions, or (c) difficulty such that trained adapters did no better than random chance. The selected tasks and datasets are detailed in Appendix F. For each task, we trained both LoRA and aLoRA adapters for 4 instruct-tuned models of various sizes. To ensure a fair comparison, for each task-model pair, using the validation set performance we performed grid search over 4 rank values and 5 learning rate values[10], and followed typical LoRA SFT best practices for all parameters. Note that we did not observe any pattern to the performance-maximizing rank value in these experiments, in particular there was not a clear pattern of LoRA or aLoRA needing higher or lower rank. The invocation sequence was set to equal the model's chat template generation prompt. See Appendix F for further details.

Results are shown in Figure 4. While there is decent variability due to the generally small size of the datasets and non-exhaustiveness of the hyperparameter search, overall it can be seen that neither aLoRA or LoRA has a consistent accuracy advantage. The median performance difference is 0.0%, and the mean performance difference is 0.6% in favor of LoRA (3.06% in favor of activated LoRA when restricted to MC tasks). Using a 2 sided t-test, the p-value for rejecting the null hypothesis of equal means is 0.8, failing to reject at level 0.05. This supports our thesis that aLoRA can achieve comparable accuracy to LoRA while significantly outperforming in inference costs.

## 4.2 Intrinsics Tasks

In this subsection, we consider real-world tasks that fit the framework of *intrinsics*, specifically, settings where the task requires a potentially long input that either (a) already has base model KV cache available from either being input to or generated by the base model, or (b) does not have (complete) base model KV cache available, but base model KV cache created through an aLoRA call will be used in a subsequent base model call. We draw from those proposed in [5] for a RAG pipeline, though these intrinsics are not limited to RAG. See [5] for deeper motivation for each intrinsic and extensive experimental results comparing their LoRA adapters to strong baselines.

In [5], LoRA adapters were trained for the intrinsics using the Granite 3.2 8b Instruct model. For direct comparison to their released adapters, we train corresponding aLoRA adapters (using the same datasets and chat-template-based formatting) and compare their performance in terms of generation

---

[9]PEFT contains the model implementation and handles inference. Training in our experiments is done using a standard Huggingface TRL [38] trainer (SFTTrainer).

[10]Note that this experiment represents 1120 separate adapter training runs.

quality and/or accuracy.[11] Note that since all the below models use the same base model, they can be swapped in and out as needed in *the same flow*. The reader can envision the wide range of possibilities enabled by these intrinsics, e.g. for RAG. See the Appendix G for additional details for this section.

**Uncertainty Quantification:** This intrinsic provides a Certainty score for model responses to user questions. The model will respond with a number from 0 to 9, corresponding to $5\%, 15\%, 25\%, \ldots, 95\%$ confidence respectively. Training data for these confidence scores are obtained by applying the UQ calibration method of [32] to a large, diverse set of benchmark question answering datasets, quantizing the predicted confidences and using these predicted confidences as SFT targets for the adapter. Following the chat template, the invocation sequence is 4 tokens: `<|start_of_role|>certainty<|end_of_role|>`. Note that the model is evaluating responses from its base model - in other words, it cannot be applied to generations from other models. The aLoRA architecture is thus particularly well-suited to this use case. In practice, the goal is to provide a highly efficient uncertainty score without having to resort to expensive larger judge models.

| Certainty Score | LoRA | aLoRA |
|---|---|---|
| **MAE** | 0.50 | 0.49 |

The Uncertainty Quantification intrinsic returns an ordinal score, so we compute the mean absolute error between the predicted integer and the target integer in the SFT data. Results are shown in Figure 5. Note that performance is largely unchanged using aLoRA instead of standard LoRA.

Figure 5: Test error for the Uncertainty Quantification intrinsic. Note that aLoRA does not lose meaningful performance.

**Answerability Determination:** This intrinsic assesses whether a user's final query in a multi-turn conversation can be answered given the retrieved documents. In RAG or other settings, this decides whether to proceed with generation or abstain with an "I don't know" response. The invocation sequence is set to `<|start_of_role|>answerability<|end_of_role|>`. We tested on binary answerability classification on the single-turn SQUADRun Benchmark [30] with the user query and the supporting documents, and the multi-turn MT-RAG Benchmark [15] using full multi-turn conversation history along with the supporting documents. Figure 6 shows the results. Overall, the aLoRA model does not lose performance relative to the LoRA model.

**Query Rewrite:** This intrinsic is generally applicable for multi-turn conversational use cases, and its role is to perform rewrites of user queries for better performance for the downstream tasks. It is especially useful in RAG settings, see the metrics and evaluation results below. The query rewrite task is as follows: given a multi-turn conversation between a user and an AI assistant, de-contextualize the last user utterance (query) by rewriting it (whenever necessary) into an equivalent version that is standalone and can be understood by itself. The rewritten query can be sent to downstream tasks (e.g., to a retriever in a RAG setting) as a better replacement for the original user query, and without the need for (a potentially very long) context.

| Dataset | Adapter | Unans. | | | Ans. | | | Weighted F1 |
|---|---|---|---|---|---|---|---|---|
| | | P | R | F1 | P | R | F1 | |
| SQUADRUN Dev | LoRA | 84.2 | 68.0 | 75.2 | 73.1 | 87.2 | 79.5 | 77.4 |
| | aLoRA | 83.0 | 81.1 | 82.0 | 81.4 | 83.3 | 82.4 | 82.2 |
| MT-RAG Benchmark | LoRA | 85.4 | 89.3 | 87.3 | 87.0 | 82.4 | 84.6 | 86.1 |
| | aLoRA | 85.8 | 89.1 | 87.4 | 86.8 | 83.0 | 84.9 | 86.2 |

Figure 6: Comparison of classification performance of LoRA vs. aLoRA across the SQUADRUN Dev set and MT-RAG benchmark. Metrics are broken down by class (Unanswerable vs. Answerable) and include precision (P), recall (R), F1 score, and weighted F1.

Retrieval recall evaluation (Recall@k) with different query rewrite strategies, evaluated on full, non-standalone and standalone subsets of MT-RAG dataset [15] are shown in Figure 7a. All retrieved passages are obtained using the Elser retriever with the same settings as in the above paper. We evaluate on three different subsets of MT-RAG detailed in the appendix.

Answer quality evaluation using RAGAS Faithfulness (RAGAS-F) and RAD-Bench on full, non-standalone and standalone subsets of MT-RAG dataset are shown in Table 7b (see appendix for details). Note that throughout, performance numbers for aLoRA and LoRA are within a point or two. See [5] for additional comparisons showing that these approaches outperform benchmarks and are very close to the performance with gold rewrites.

---

[11]As they released LoRA adapters for only one base model, we limit ourselves to their choice of base model.

| | Full MT-RAG | | | Non-standalone | | | Standalone | | |
|---|---|---|---|---|---|---|---|---|---|
| **Strategy** | **R@5** | **R@10** | **R@20** | **R@5** | **R@10** | **R@20** | **R@5** | **R@10** | **R@20** |
| aLoRA | 0.54 | 0.66 | 0.74 | 0.42 | 0.54 | 0.64 | 0.63 | 0.75 | 0.82 |
| LoRA | 0.56 | 0.68 | 0.76 | 0.44 | 0.57 | 0.66 | 0.63 | 0.75 | 0.83 |

(a) Retrieval (Recall@5, @10, and @20)

| | Full MT-RAG | | Non-standalone | | Standalone | |
|---|---|---|---|---|---|---|
| **Strategy** | **RAGAS-F** | **RAD-Bench** | **RAGAS-F** | **RAD-Bench** | **RAGAS-F** | **RAD-Bench** |
| aLoRA | 0.81 | 0.69 | 0.77 | 0.69 | 0.83 | 0.70 |
| LoRA | 0.81 | 0.70 | 0.79 | 0.69 | 0.83 | 0.71 |

(b) Answer generation quality (RAGAS-F, RAD-Bench)

Figure 7: Impact of query rewrite strategies on both retrieval and generation tasks across subsets of MT-RAG.

**Jailbreak Detection:** This intrinsic is designed for detecting jailbreak risk within user prompts. Prompts with jailbreak risk vary across a wide range of attack styles - from direct instructions, to encoding-style, social-hacking based attacks and even ones that exploit special token or context overload [31]. In our experiments we focused on training intrinsics for detecting social hacking style of adversarial prompts. As with prior intrinsics, the aLoRA/LoRA detectors are trained in the same conditions with a rank of 32. The intrinsic is trained to return a binary label - "Y" indicating jailbreak risk present and "N" indicating no risk.

| | **Acc** | **TPR** | **FPR** |
|---|---|---|---|
| aLoRA | 0.925 | 0.863 | 0.013 |
| LoRA | 0.943 | 0.898 | 0.011 |

Figure 8: Performance of jailbreak risk detectors.

After training, we evaluate the jailbreak intrinsic on new out-of-distribution datasets not used in training to robustly assess generalizability [46]. The out-of-distribution datasets comprise of 3,282 samples containing a mixture of samples with jailbreak risk [33, 20, 39] and benign samples [4]. As with the other intrinsics, we see very little performance difference between LoRA and aLoRA.

## 5 Conclusion

We presented Activated LoRA (aLoRA), a novel modification of the LoRA framework enabling efficient and dynamic adaptation of large language models without requiring recomputation of the key-value (KV) cache. By modifying LoRA to adapt weights only for tokens generated after activation, aLoRA facilitates seamless integration into multiturn settings, enabling the use of specialized "intrinsics" for well-defined operations within a broader conversational or processing pipeline. While aLoRA adapters sometimes require higher rank $r$ than LoRA, this is not a meaningful downside at inference time since the rank $r$ matrix multiplications are a very small part of overall inference costs (although training costs do increase).

Our experiments demonstrate that aLoRA maintains competitive accuracy compared to standard LoRA while significantly reducing inference costs. This capability was showcased through applications in uncertainty quantification, answerability determination, hallucination detection, query rewrite, and jailbreak detection. The flexibility and efficiency of aLoRA highlight its potential to streamline the deployment of modular, task-specific models in complex workflows, paving the way for more adaptive and responsive LLM-driven systems.

We anticipate that aLoRA can be profitably applied to a vast array of intrinsics-style finetuning tasks that can be created for applications throughout the agentic, chat, and reasoning spaces. Future work will explore developing and proving these expanded use cases.

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

## A    Invocation sequences

In our implementation, the aLoRA weights are activated one token after the *start* of the invocation sequence. For the benchmark dataset experiments, we simply set the invocation sequence to be the "generation prompt" specified by the base model's chat template. For the various intrinsics experiments, we most often used a 3-4 token length sequence which included a one-word description of the task.

The invocation sequence has several benefits. The invocation sequence can be designed to

- Conform to the chat template (for instance by making the aLoRA response its own turn with a specialized role).
- Provide an (optional) short prompt to aid the learning process.
- Give the adapter weights a few more tokens to process the input prior to generating the output, often improving performance in practice.

In our current implementation, the invocation sequence denoting the starting point of adaptation is fixed, with the option to have a variable prompt follow it prior to actual generation. In principle, there is no need for any consistency of input sequences so long as the point at which the adapter weights should be turned on can be indicated by some means.

## B    LLM Transformer architecture

A generic architecture is shown in Figure 9.

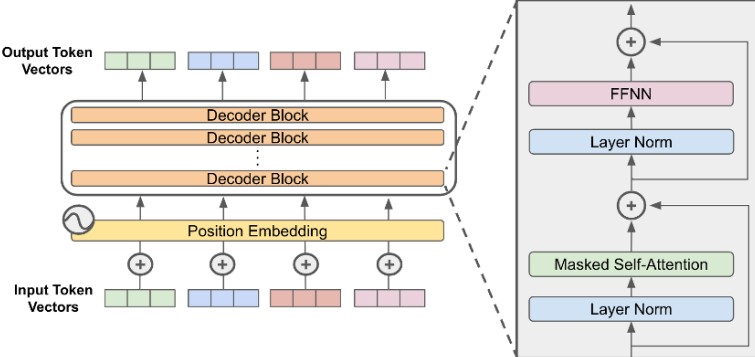

Figure 9: A generic LLM architecture.

## C    Proofs

We repeat the propositions here from the main text and prove them.

**Proposition 3** (Proposition 1 (KV equivalence and aLoRA inference)). *For the causal decoder-only transformers we consider, the keys and values (actually, all internal states) prior to $t_{\text{invoke}}$ are identical for the base model and any aLoRA adapter model using* (6). *Specifically,* $K^{\text{base}}_{1:t_{\text{invoke}}-1} = K^{\text{adapter}}_{1:t_{\text{invoke}}-1}$, *and* $V^{\text{base}}_{1:t_{\text{invoke}}-1} = V^{\text{adapter}}_{1:t_{\text{invoke}}-1}$. *Inference with the aLoRA adapted model can be done causally one token at a time (by simply increasing $t$ in* (6) *iteratively) with KV cache reuse.*

*Proof of Proposition 1 (KV equivalence and aLoRA inference).* In the proof, for notational simplicity we focus on adapters to the attention blocks, but the treatment for adapters to the MLP blocks is the same. By construction (6), for every token index $i < t_{\text{invoke}}$ the adapter's weight matrices coincide with the base model's:

$$W^Q_{\text{adapter}}\big|_i = W^Q, \quad W^K_{\text{adapter}}\big|_i = W^K, \quad W^V_{\text{adapter}}\big|_i = W^V.$$

Hence for each $i < t_{\text{invoke}}$ and starting with the first attention layer,

$$Q_i^{\text{adapter}} = x_i\big(W^Q + 0\big) = x_i W^Q = Q_i^{\text{base}},$$

and similarly

$$K_i^{\text{adapter}} = x_i W^K = K_i^{\text{base}}, \quad V_i^{\text{adapter}} = x_i W^V = V_i^{\text{base}}.$$

Since the transformer's subsequent layer outputs (including all MLP and layer-norm states) are deterministic functions of these $Q, K, V$ up to $i$, it follows by induction on layer depth that *all* internal states for tokens $1, \ldots, t_{\text{invoke}} - 1$ agree between the two models. In particular the cached key- and value-matrices satisfy

$$K_{1:t_{\text{invoke}}-1}^{\text{adapter}} = K_{1:t_{\text{invoke}}-1}^{\text{base}}, \quad V_{1:t_{\text{invoke}}-1}^{\text{adapter}} = V_{1:t_{\text{invoke}}-1}^{\text{base}}.$$

During generation at time $t \geq t_{\text{invoke}}$, both models proceed token-by-token via exactly the same causal-attention mechanism (1), merely appending the new $(Q_t, K_t, V_t)$ row and reusing the previously cached rows. Since up to $t - 1$ those rows coincide, the adapter may *reuse* the base model's KV cache for the first $t_{\text{invoke}} - 1$ tokens, and then continue to grow its own cache for tokens $t_{\text{invoke}}, \ldots, t - 1$. This establishes that aLoRA inference can indeed be performed causally, one token at a time, with full KV cache reuse. $\qquad\square$

**Proposition 4** (Proposition 2 (aLoRA vs. LoRA inference costs))**.** *Consider invoking an adapter with $T_{\text{cache}}$ tokens of input for which a base model KV cache exists and $T_{\text{new}} \ll T_{\text{cache}}$ input tokens without cache.[12] The first token generated by the aLoRA adapter requires $O(T_{\text{cache}}T_{\text{new}})$ operations, while LoRA requires $O((T_{\text{cache}} + T_{\text{new}})^2)$. Furthermore, aLoRA must maintain only $O(T_{\text{new}})$ additional KV cache memory, while LoRA requires additional $O(T_{\text{cache}} + T_{\text{new}})$. If $N$ distinct adapters share the first $T_{\text{cache}}$ tokens as input, aLoRA costs become $O(NT_{\text{cache}}T_{\text{new}})$ and $O(NT_{\text{new}})$, while LoRA has $O(N(T_{\text{cache}} + T_{\text{new}})^2)$ and $O(N(T_{\text{cache}} + T_{\text{new}}))$.*

*Proof of Proposition 2 (aLoRA vs. LoRA inference costs).* Let the input be partitioned into $T_{\text{cache}}$ tokens whose base-model cache is already available, and $T_{\text{new}}$ tokens on which the adapter must act without precomputed cache. We compare the cost of generating the first new token:

- **aLoRA.** To generate token $t = T_{\text{cache}} + T_{\text{new}} + 1$, we do a forward pass of the transformer, where we have keys and values for the first $T_{\text{cache}}$ tokens (and therefore do not need to recompute these). The MLP blocks do not interact between tokens, so scale linearly only with $T_{\text{new}} + 1$, hidden dimension, and number of layers. For each attention layer, we
  1. compute $T_{\text{new}} + 1$ new rows each of $Q, K, V$ in $O(T_{\text{new}}d_k)$,
  2. form the attention scores by multiplying these new queries against the cached-plus-new key-matrix of size $(T_{\text{cache}} + T_{\text{new}}) \times d_k$, costing $O((T_{\text{cache}} + T_{\text{new}})\,d_k)$,
  3. apply the $T_{\text{new}} + 1$ masked softmaxes and weighted sums over $(T_{\text{cache}} + T_{\text{new}})$ values in $O(T_{\text{new}}(T_{\text{cache}} + T_{\text{new}})\,d_v)$,
  4. form the adapter's own new cache entries, yielding at most $O(T_{\text{new}})$ extra storage.

  Hence, considering hidden dimension and number of layers as constant, the dominant cost is $O(T_{\text{cache}} + T_{\text{new}}(T_{\text{new}} + T_{\text{cache}})) = O(T_{\text{new}}T_{\text{cache}})$, and extra memory $O(T_{\text{new}})$.

- **LoRA.** Since LoRA's rank-$r$ updates apply to *all* tokens—including those in the original context—no prefill cache may be reused. Generating the first token thus requires recomputing attention over $(T_{\text{cache}} + T_{\text{new}})$ tokens, incurring $O\big((T_{\text{cache}} + T_{\text{new}})^2\big) = O\big(T_{\text{cache}}^2\big) \gg O(T_{\text{new}}T_{\text{cache}})$ compute, and storing $O(T_{\text{cache}} + T_{\text{new}})$ key/value rows.

If $N$ distinct adapters each process the same $T_{\text{cache}} + T_{\text{new}}$ context but disjoint $T_{\text{new}}$ segments, the aLoRA per-adapter cost scales as $O(T_{\text{cache}} + N\,T_{\text{new}}T_{\text{cache}})$ time and $O(N\,T_{\text{new}})$ memory, whereas LoRA's cost scales on the full quadratic context for each adapter (totaling $O\big(N\,(T_{\text{cache}} + T_{\text{new}})^2\big)$ time and $O\big(N\,(T_{\text{cache}} + T_{\text{new}})\big)$ memory). $\qquad\square$

---

[12]We presume that $T_{\text{new}}$ is the invocation sequence and following.

## D  Adapter Model Capacity and Increased Rank

Figure 10 illustrates this observation (described in the main text). In our experiments, rank of 32 seems to be sufficient in most cases, which is still vastly smaller than the size of the base model. Whenever comparing to LoRA models, we chose a LoRA rank that achieved top performance for LoRA, rather than attempting to match the ranks between LoRA and aLoRA.

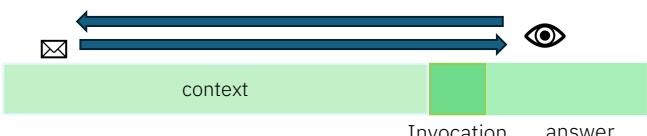

(a) LoRA is able to modify the keys (K) of task-relevant values (V) in the context attention layers for the modified queries (Q) recover via the attention mechanism.

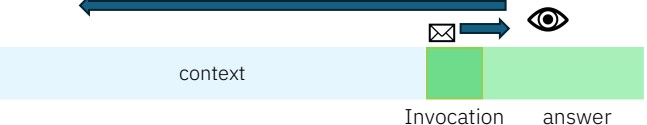

(b) For the context, aLoRA is not able to modify context KVs, it must rely on learning modified Qs alone to extract information from existing context KVs.

Figure 10: Intuition for why aLoRA adapters often require increased rank $r$. ALoRA is not able to modify the keys and values of input tokens prior to the invocation sequence to send new "messages" forward to the generation step, and a too small rank constraint limits its capacity to do this.

## E  Inference Timing Performance Evaluation Details

In this section we document the methodology followed in the experiments leading to Figure 3. These experiments are modeled, particularly for the smaller prompt lengths (10k tokens) after multi-turn RAG situations where a collection of documents concatenated with multiple conversation turns are followed by an answer of total length a few hundred tokens, and then where judges are used to determine various qualities of the answer.

For each model considered, 5 low rank adapters with random weights were created for ranks 8 and 32, and independent vLLM instances were launched for each model and collection of adapters in two modalities: a server where the adapters are regarded as regular LoRAs (where we used rank 8) and a server where they are regarded as activated LoRAs (where we used rank 32). We note that while in our experiments such high rank discrepancies are the exception rather than the rule, we adopted this setup so as to give an additional potential advantage to LoRA in the evaluation, as a lower rank update is in principle cheaper to compute; we also remark that we saw very little change in the experiments compared to a settign where both ranks match. For each model, the KV cache size in tokens as calculated by vLLM is used to compute the batch size to use by dividing the KV cache size by the length of the sequence that the LLM will be tested on (including initial prompt, generation and evaluation). A total of three batches are tested consecutively for each combination of model, number of evaluators and prompt length. Each batch is processed in sequential stages: answer generation for the entire batch is then followed by one or more evaluations, also done for the entire batch.

Batches are created by choosing prompts at random while ensuring that their length, after tokenization using any given model's tokenizer, is precisely the desired length. The length of the answers (256 tokens) and evaluations (16 tokens) are enforced by passing to vLLM's generate method matching minimum and maximum number of tokens to generate.

# F  Benchmark SFT experiment details

## F.1  Models

Llama 3.2 1B Instruct (`meta-llama/Llama-3.2-1B-Instruct` on Hug-gingface), Llama 3.2 3B Instruct (`meta-llama/Llama-3.2-3B-Instruct`), Llama3.1 8B (`meta-llama/Llama-3.1-8B-Instruct`) [9], and Mistral 7B (`mistralai/Mistral-7B-Instruct-v0.3`) [35].

## F.2  Tasks and Datasets

Here we provide additional information on the SFT tasks in Figure 4. Table 1 gives information for each dataset on the size of the train/validation/test splits, as well as the number of multiple-choice responses for the multiple-choice tasks. Freeform tasks have unstructured natural language strings as target output.

Table 2 provides the Huggingface path for each dataset. The URL can be recovered as `https://huggingface.co/datasets/Lots-of-LoRAs/PATH` where `PATH` is the name indicated in Table 2.

We next provide the task definition prompt for each of the datasets. Note that the actual datasets include this task definition, in-context-learning (ICL) examples, and the test input as part of the full prompt to the LLM. Examples can be seen on the Huggingface pages for each dataset.

**Bengali Hate Speech Classification**  "In this task, you are given a hateful post in Bengali that expresses hate or encourages violence towards a person or a group based on the protected characteristics such as race, religion, sex, and sexual orientation. You are expected to classify the post into four classes: Religious, Political, Geopolitical and Personal depending on the topic."

**WIQA: Effect Classification**  "In this task you will be given a process, and a question. The process contains a sequence of steps that happen in order. The question asks about the effect of a certain event on another event. If the first event has a positive effect on the second event, answer with "for", if it has a negative effect, answer with "against". If there's no causal relationship between the two, answer with "none"."

**MMLU Conceptual Physics MCQA**  "You are given a question on conceptual physics. You are also given 4 answer options (associated with "A", "B", "C", "D"), out of which only one is correct. You need to answer the question by selecting the correct option. You should only answer with the choice letter, not the whole answer."

**MMLU College Computer Science MCQA**  "You are given a question on college computer science. You are also given 4 answer options (associated with "A", "B", "C", "D"), out of which only one is correct. You need to answer the question by selecting the correct option. You should only answer with the choice letter, not the whole answer."

**SocialIQA Question Generation**  "In this task, you're given context and an answer. Your task is to generate the question for this answer based on the given context with commonsense reasoning about social situations."

**Hindi Sentence Perturbation**  "Given a sentence in Hindi, generate a new Hindi sentence by performing small changes on the sentence. Here, make sure that the changes are semantically related and syntactically similar to the input. And the generated sentence should have high commonsense plausibility, that is to have reasonable probability of it being true."

**SuperGLUE Question Generation**  "In this task, you are given Wikipedia articles on a range of topics, we ask you to write a question based on the content of the articles that can be answered in a binary manner i.e. True or False."

| Type | Task | #Options | Train | Valid | Test |
|---|---|---|---|---|---|
| Multiple Choice | Bengali Hate Speech Classification | 3 | 18.1k | 227 | 227 |
| | WIQA: Effect Classification | 3 | 5.2k | 650 | 650 |
| | MMLU Conceptual Physics MCQA | 4 | 142 | 18 | 18 |
| | MMLU College Computer Science MCQA | 4 | 90 | 12 | 11 |
| Freeform | SocialIQA Question Generation | | 5.2k | 650 | 650 |
| | Hindi Sentence Perturbation | | 5.18k | 648 | 647 |
| | SuperGLUE Question Generation | | 1.51k | 190 | 189 |

Table 1: Dataset sizes and option counts for selected tasks.

| Task | Name on `Lots-of-LoRAs` (Huggingface) |
|---|---|
| Bengali Hate Speech Classification | `task1494_bengali_hate_speech_classification` |
| WIQA: Effect Classification | `task1727_wiqa_what_is_the_effect` |
| MMLU Conceptual Physics | `task693_mmmlu_answer_generation_conceptual_physics` |
| MMLU College Computer Science | `task688_mmmlu_answer_generation_college_computer_science` |
| SocialIQA Question Generation | `task581_socialiqa_question_generation` |
| Hindi Sentence Perturbation | `task407_mickey_hi_sentence_perturbation_generation` |
| SuperGLUE Question Generation | `task1660_super_glue_question_generation` |

Table 2: Task types and their Hugging Face dataset paths on `Lots-of-LoRAs`.

### F.3 Training and evaluation details

Following typical LoRA SFT best practices, for both LoRA and aLoRA we used 4 training epochs, alpha of 32, dropout of 0.05, adapted the K, Q, and V modules in all layers, and searched over ranks $[6, 8, 16, 32]$ and learning rates $[3 \times 10^{-6}, 10^{-5}, 3 \times 10^{-5}, 10^{-4}, 3 \times 10^{-4}]$. Batch size of 8 was used, with 16-bit arithmetic precision. Hyperparameters were selected by taking the configuration that performed best on the validation set, and reported performance was computed on the test set for those selected models. All training runs were done on single H100 GPUs.

All models were evaluated by comparing the generated answers to golden answers provided in the dataset[13], and percent correctness was used as the metric. For the multiple-choice tasks, agreement was simple to evaluate. For the freeform tasks, an LLM judge prompt compared the generated answers to the golden answers and output a binary decision.

## G  Additional details for intrinsics experiments

For the intrinsics tasks, all attention weights (keys, queries, values) were adapted in all layers, using rank 32 adapters. The learning rate and number of epochs were tuned to achieve the best validation performance (as was the case for the LoRA adapters of [5]). Intrinsics training tasks each used an 8 GPU H100 node.

### G.1  Uncertainty Quantification

**Certainty score interpretation** The returned percentages are *calibrated* in the following sense: given a set of answers assigned a certainty score of X%, approximately X% of these answers should be correct. Here "approximately" can be quantified via the expected calibration error, or ECE. Essentially what happens is teaching the adapter model what the base model knows and doesn't know. This inherently requires generalization to questions of wildly varying difficulty (some of which may be trick questions!) and to settings not in training. Intuitively, it does this by extrapolating based on related questions it has been evaluated on in training - this is an inherently inexact process and leads to some hedging. **Training pipeline** First, a probe-based model was trained to produce calibrated certainty scores, using a large diverse collection of QA datasets detailed in Appendix H.1. Note that throughout, the chat template was used. The procedure for this was as follows:

---

[13]Note that some datasets may contain some label noise.

1. A "meta-dataset" was created containing User inputs, Answer generations (from the base model), and correctness labels for those generations.

2. For each row in the meta-datset, the base model was prompted with input of the form (User inputs, Answer generations, meta prompt), where the meta prompt was

   ```
   Is the above answer correct?\n <A> Yes, \n No, \nAnswer:
   ```

   and one token was generated.

3. The hidden state from the last layer of the model was saved off for the generated token. This was then combined with the correctness labels from step (1) to create a dataset of (hidden states, correctness labels).

4. A 3 layer MLP was trained on the dataset of the previous step. This is known in the literature as a probe.

5. The logits of the output of this MLP on held-out validation datasets were converted into probabilities, and the ECE was computed.

6. Temperature scaling was applied here to minimize the ECE, resulting in test dataset ECE of 0.02.

The above follows the procedure of [32] for freeform responses, and was applied to both the multiple choice and freeform data for consistency.

Having a calibrated probe model, we then created a teacher dataset, where all datasets were processed by the probe model and the computed probabilities were recorded and quantized in steps of 10% (05% to 95%). This teacher dataset served as the training data for the aLoRA model, which was trained to use the invocation sequence

```
<|start_of_role|>certainty<|end_of_role|>
```

and to generate the quantized percentage values.

## G.2  Answerability determination

The input to the model is a list of conversational turns and a list of documents converted to a string using `apply_chat_template` function. These turns can alternate between the user and assistant roles. The last turn is from the user. The list of documents is a dictionary with text field, which contains the text of the corresponding document. To prompt the aLoRA adapter to determine answerability, a special answerability role is used to trigger this capability of the model. The role includes the keyword `"answerability"`: `<|start_of_role|>answerability<|end_of_role|>` When prompted with the above input, the model generates the answerable or unanswerable output. See [5] for more details.

**Training Details** The aLoRA and LoRA adapters were fine-tuned under the following regime: rank = 32, learning rate = 5e-6, number of epochs = 25, with early stopping based on validation set, and 90/10 split between training and validation.

## G.3  Query Rewrite

**Usage** The input to the model is a list of conversational turns converted to a string using `apply_chat_template` function. These turns can alternate between the user and assistant roles, and the last turn is assumed to be from the user. To prompt the aLoRA adapter to rewrite the last user turn, a special rewrite role is used to trigger the rewrite capability of the model. The role includes the keyword "rewrite" followed by a short description of the query rewrite task. Even though one main application for query rewrite is in RAG settings, this intrinsic can be used to rewrite user questions for other conversational use cases (e.g., to access a database, or other APIs, or tools). As such, the adapter does not need any RAG documents (that may be present in the context, in a RAG setting) and uses only the dialog turns with what is being said between the user and assistant.

**Training** Both the aLoRA and LoRA adapters were fine-tuned under the following regime: rank = 32, number of epochs = 25, with early stopping based on validation set, and 90/10 split between training and validation.

**Evaluation data** We evaluate on three different subsets of MT-RAG: a) full MT-RAG dataset (842 data points with last user turns); b) the non-standalone subset of MT-RAG dataset, which is a subset of 260 (out of 842) last user turns that were annotated by humans as non-standalone (i.e., they are dependent on the prior context); c) the standalone subset of MT-RAG dataset, which is the complementary subset, with all the last user turns that were annotated by humans as standalone.

**Answer generation quality** We also evaluate answer generation quality, with top-k passages retrieved under the various query rewrite strategies for the retriever. We choose here $k = 20$, but similar trends take place for other values of $k$. We used Granite-3.2-8b instruct as the answer generator, and RAGAS Faithfulness (RAGAS-F) and RAD-Bench score as metrics for answer quality. We use the same three testsets as above.

### G.4 Jailbreak Detection

**Usage** The input to the model is a single prompt to assess for harmful content. Currently, the intrinsics operate on single turn queries. To prompt the aLoRA/LoRA adapters `<|start_of_role|>jailbreak<|end_of_role|>` is used.

**Training** Both the aLoRA and LoRA adapters were fine-tuned under the following regime: rank = 32, fixed 5,000 optimization steps, 6e-5 learning rate with the Adam optimizer.

## H Training datasets for intrinsics

### H.1 QA datasets for Uncertainty Quantification Intrinsic

The following datasets were used for calibration and/or finetuning.

- BigBench
- MRQA
- newsqa
- trivia_qa
- search_qa
- openbookqa
- web_questions
- smiles-qa
- orca-math
- ARC-Easy
- commonsense_qa
- social_i_qa
- super_glue
- figqa
- riddle_sense
- ag_news
- medmcqa
- dream
- codah
- piqa

## H.2 Training data for Query Rewrite Intrinsic

The training data contains both: 1) standalone examples, which teach the adapter to refrain from rewriting user questions that are already standalone, and 2) non-standalone examples containing a diversity of patterns that are used to teach the adapter to expand the user turn so that it becomes standalone.

The training data used the publicly available Cloud corpus of technical documentation pages from MT-RAG.[14] Based on this corpus of documents, a dataset was created consisting of high-quality, human-created conversations, where the last turn of the conversation comes into versions: non-standalone version, and corresponding standalone version.

## H.3 Training data for Answerability Determination Intrinsic

The training data uses the publicly available Government corpus from MT-RAG [15] as the source of documents. Based on this corpus, the dataset consists of a mix of human-created and synthetically generated multi-turn conversations. It includes two types of examples: (1) Answerable queries, where the final user question can be answered based on the provided documents. These examples teach the adapter to recognize when sufficient information is present to support an answer. (2) Unanswerable queries, where the documents lack the necessary information to answer the final user query. Mixtral was used as an automatic judge to validate the answerability labels and filter out noisy samples.

## H.4 Training data for Jailbreak Intrinsic

The Jailbreak Intrinsic was trained on 40,000 harmful and benign samples. This is composed of several sub-sampled open source datasets (Gandalf Ignore Instructions [27], Awesome ChatGPT Prompts [1], BoolQ [3], SAP [6], UltraChat [8], super natural instructions [42]) as well as non-public prompt datasets of harmful/benign content.

---

[14]https://github.com/IBM/mt-rag-benchmark

