# OpenReview forum: "Activated LoRA: Fine-tuned LLMs for Intrinsics"
_NeurIPS.cc/2025/Conference — NeurIPS 2025 poster_

### Official Review · Reviewer_mmG2 · 2025-06-28

**Clarity:** 3
**Significance:** 4
**Originality:** 4
**Rating:** 5
**Confidence:** 4

**Summary:**

This paper introduces Activated LoRA (aLoRA), a novel modification of the LoRA framework designed to improve inference efficiency in dynamic, multi-turn settings. The key problem addressed is the significant computational overhead incurred when switching between standard LoRA adapters, which requires a full recomputation of the KV cache for the entire context history . aLoRA solves this by adapting model weights only for tokens that appear after a specific invocation point in the sequence. This allows the aLoRA adapter to directly reuse the KV cache generated by the base model for all preceding context, thereby eliminating the need for recalculation and enabling instantaneous switching. The authors frame these specialized, on-demand adapters as "intrinsics". Through extensive experiments, the paper demonstrates that aLoRA achieves substantial inference speedups while maintaining competitive accuracy compared to standard LoRA across a variety of tasks.

**Questions:**

The paper effectively demonstrates the utility of aLoRA within a "late prompting" framework. However, it is unclear how this method would handle more complex prompts where instructions and content are interleaved. For example, consider the following prompt:

    `Here is my article, and I need you to polish it... [Article content]... Please use the following evaluation criteria when polishing... [Criteria content]... Now, based on the criteria above, please optimize my article sentence by sentence and provide your reasoning.`


In a scenario like this, which does not fit a simple content -> instruction model, where would the single invocation_sequence be placed to activate the desired aLoRA? Could the authors provide an example or elaborate on the strategy for applying aLoRA in these more complex, mixed-prompt situations?

**Ethical Concerns:**

["NO or VERY MINOR ethics concerns only"]

**Final Justification:**

The authors' responses addressed my concerns regarding the external controller and prompt format. I believe their responses have deepened my understanding of the article. I have no further concerns. I believe this article has practical application value, the experiments are solide, and it can be accepted by the main conference.

**Limitations:**

yes

**Quality:**

3

**Strengths And Weaknesses:**

### Strengths

1.  Focus on real-world problems: This paper explores a key bottleneck in real-world deployments of LLM. The inefficiency of switching LoRA adapters in a agentic pipeline or multi-turn chat is an important real-world problem.
2.  Effective and well-designed method: The proposed aLoRA method provides an effective and elegant solution to the above problems. The core idea is to reuse the key-value cache of the base model by delaying the activation of the adapter weights, which is intuitive and can directly improve efficiency.
3.  Comprehensive experimental validation: The paper's claims are supported by comprehensive experiments. The authors provide convincing evidence of inference speed improvements for different models, demonstrating real-world advantages over  vLLM engines. The experiments show that the efficiency gains do not come at the expense of performance, and aLoRA remains competitive with standard LoRA.

### Weaknesses

1.  Reliance on an external judge for activation: The aLoRA framework relies on a "invocation token sequence" that is "often invoked programmatically" to signal the activation point. This implies the existence of an external controller or “judge” that knows precisely when to call a specific intrinsic function. In many real-world unstructured conversational environments, determining the right timing and appropriate intrinsic function call is a complex task in itself. This paper assumes that such external control logic is readily available, which may limit its applicability in scenarios that lack predefined operational flows.
2.  Oversimplified Assumption of Prompt Structure: The paper's motivation heavily relies on a simplified "late prompting" (content -> instruction) paradigm to highlight the benefits of cache reuse. However, practical user prompts are often more complex, with interleaved instructions and content.  This oversimplification may pose challenges when applying the method outside of well-defined, structured tasks.

---

> ### Author Rebuttal · Authors · 2025-07-30
>
> We thank the reviewer for their insightful questions and positive evaluation of our work, highlighting the real-world relevance and design of our method and convincingness of our experiments. We respond to questions below.
>
> > ”The aLoRA framework relies on a "invocation token sequence"…which may limit its applicability in scenarios that lack predefined operational flows.”
>
> This actually has a very nice solution for the case where there is no external controller. While beyond the scope of the present work, this invocation is not hard to add to the tool-calling ability of modern LLMs by listing it to the LLM as an available tool it can call.
>
> That said, agentic settings can always have a controller available in principle, as by definition the turns of generation are being driven by the system, not the user. Further, the existence of a controller does not mean that the flow is locked-in, any relevant information for the controller can be created bespoke by an LLM. We are intensely motivated by this agentic setting as we believe it will explode in relevance over the coming years and become more important than the classic unstructured conversational setting.
>
> > ”Assumption of Prompt Structure: The paper's motivation heavily relies on a simplified "late prompting" (content -> instruction) paradigm to highlight the benefits of cache reuse. However, practical user prompts are often more complex, with interleaved instructions and content.”
>
> Fortunately, this imagined scenario won’t be a limitation. See our answer to your question below.
>
> > ”consider the following prompt: Here is my article, and I need you to polish it... [Article content]... Please use the following evaluation criteria when polishing... [Criteria content]... Now, based on the criteria above, please optimize my article sentence by sentence and provide your reasoning.”
>
> Thank you for this question! We would like to clarify that our concept of “late prompting” is one abstraction level up from this example, we are completely agnostic to how the user chooses to present their request. Indeed, the scenario you have written exactly follows the “late prompting” framework we propose! Firstly, note the above is a single prompt. For this case, as-is there is only one model call, so no redundant prefilling, and no issues – this is neither late nor early prompting in our framework. You could imagine a follow up check to make sure the criteria are followed (either programmatic, or by the user). Here, in fact, typical user behavior is late prompting! Specifically, the user will have a follow up question “Did you follow all criteria carefully” (which is late prompting as the original generation is in context prior to this statement), followed by "given the criteria that were missed, can you rewrite the content". They will almost never follow the early-prompting scenario, which would be to copy-paste the prior generated article back into the context window following the “Did you follow all criteria carefully”, despite the fact that the generated context is already there on the screen as a prior generation. This showcases the point of our early/late prompting framework – it is essentially a call for agentic systems to adhere more to the human-like multi-turn chat, and abandon an observed tendency to format machine-created prompts without regard for cache reuse.
>
> **Summary**
>
> We hope the above addresses your questions! We are happy to continue the discussion in the next phase.

---

> > ### Comment · Reviewer_mmG2 · 2025-08-01
> > **Thanks to the author for the reply**
> >
> > I have read the author's response, which addresses most of my concerns. I will keep my score unchanged.

---

### Official Review · Reviewer_Sg2F · 2025-07-02

**Clarity:** 2
**Significance:** 2
**Originality:** 2
**Rating:** 3
**Confidence:** 3

**Summary:**

The authors propose aLoRA, a method for reusing the key-value (KV) cache from the base model when generating tokens with LoRA adapters for specific tasks. In this approach, the KV cache is computed once using the base model for the input context and then reused across multiple LoRA adapters during generation. The paper presents both computational savings and performance evaluations to support the effectiveness of aLoRA.

**Questions:**

* If the aLoRA generation diverges substantially from the base model — for instance, adapting an English generation model to generate text in French — is this adaptation achievable?
* Section 4.1 may be misleading, as it does not directly assess the performance implications of reusing the base model’s KV cache when aLoRA is applied multiple times. It seems obvious that aLoRA can be fine-tuned for a single task without sharing, simply leveraging the base model’s KV cache — similar to how prefix-LM methods work. Hence, this result alone does not fully demonstrate the claimed benefit.

**Ethical Concerns:**

["NO or VERY MINOR ethics concerns only"]

**Final Justification:**

The authors’ discussion did not resolve my concerns regarding the novelty of this work. I am therefore maintaining my original borderline-reject score.

**Limitations:**

Yes, although it doesn’t go into enough depth.

**Paper Formatting Concerns:**

No.

**Quality:**

2

**Strengths And Weaknesses:**

**Strengths**
* Significance: For certain applications, reusing the base model’s KV cache can have clear practical benefits.

**Weaknesses**
* Quality: The analysis of LoRA adapters lacks depth, leaving gaps in understanding how adapter differences (such as to the different tasks) affect the overall performance. Also, some propositions are fairly obvious and offer limited practical insight.
* Significance: The scenario presented in the paper — using aLoRA to evaluate multiple aspects of a prompt and answer — seems rather limited. It raises the question of why the same model couldn’t generate multiple evaluations without needing separate adapters.
* Originality: The extension of KV cache reuse to multiple LoRA adapters is an interesting addition; however sharing the KV cache between models or layers is not entirely new.

---

> ### Author Rebuttal · Authors · 2025-07-30
>
> We thank the reviewer for their review, and address questions below.
>
> > ”Quality: The analysis of LoRA adapters lacks depth”
>
> Unfortunately, we don’t see anything concrete in this point that we can respond to. Please be specific regarding any gaps in analysis that you would like us to address.
>
> > ”…some propositions are fairly obvious and offer limited practical insight.”
>
> We include our propositions to formalize our claims. Their inclusion gives key practical insight into the quantitative nature of aLoRA’s advantages. We make no claims that they were “difficult” to derive and nowhere in the paper do we claim these are major contributions or even “theorems”.
>
> > ”The scenario presented in the paper — using aLoRA to evaluate multiple aspects of a prompt and answer — seems rather limited.”
>
> There seems to be some misunderstandings here which we clarify. In particular, the “evaluation” use case is by no means a limitation, but instead a strong motivating example. Specifically,
>
> 1. We note that the article includes uses for aLoRA that extend beyond evaluation; for example, the query_rewrite intrinsic is focused on full query generation, rather than only evaluation, and many of the benchmark SFT tasks are freeform. Indeed, a common use case in agents would be families of “refinement” steps where initial generations are improved and made to satisfy requirements.  In general, any well-defined task which makes sense to finetune for and will be used in concert with the base model is a potential use case for aLoRA. We will add the above discussion to the manuscript.
>
> 2. The choice of the “evaluation” use case is motivated precisely because this regime is not niche but rather, the bedrock of the entire burgeoning field of LLM agents, and to a lesser extent, certain test-time-scaling reasoning pipelines. For the importance of “evaluation” and “refinement” steps in agents, see Huang, X., Liu, W., Chen, X., Wang, X., Wang, H., Lian, D., ... & Chen, E. (2024). Understanding the planning of LLM agents: A survey. arXiv preprint arXiv:2402.02716 and many others.
>
> 3. We do not require multiple evaluators/intrinsics before we see meaningful gains. Note that for single evaluation settings, we do demonstrate double digit relative improvements in the cost of the evaluation task.
>
> > ”why the same model couldn’t generate multiple evaluations”
>
> We definitely don’t require each aLoRA adapter to only implement one evaluation – the reason to limit the number of evaluations each adapter does is not a technical limitation (though multi-task adapters are harder to train), but actually modularity for user flexibility allowing for mixing and matching of skills. Either way, the significant runtime advantages of aLoRA over LoRA remain. We view this modularity-with-almost-no-downside as a selling point of aLoRA.
>
> > ”Originality: The extension of KV cache reuse to multiple LoRA adapters is an interesting addition; however sharing the KV cache between models or layers is not entirely new.”
>
> It is unclear to us what your concern is here. If you have other references we should cite, please let us know and we will add them to the revision.
>
> > ”adapting an English generation model to generate text in French — is this adaptation achievable?”
>
> Indeed, see the SFT task results for Hindi and Bengali. Mistral 7B was not finetuned on these languages, yet aLoRA does very well.
>
> > ”Section 4.1 may be misleading, as it does not directly assess the performance implications of reusing the base model’s KV cache when aLoRA is applied multiple times.”
>
> Section 4.1 is not misleading, it does indeed directly assess the performance (accuracy) implications of reusing the base model's KV cache when aLoRA is applied multiple times. Importantly, the aLoRA architecture is extremely well-positioned for modular finetuning, i.e. multiple adapters can be tuned independently yet used together. In particular, in Section 4 we provide a collection of adapters that are meant to work together in agentic RAG scenarios, as these address pre-search, post-search and post-generation stages in RAG. Please clarify any other concerns you have here.
>
> > Limitations
>
> We will add a more explicit statement in the revision.
>
> **Summary**
>
> We hope that these clarifications and forthcoming additions will help address your concerns and merit a reconsideration of your evaluation. If concerns remain, please let us know during the discussion period!

---

> ### Comment · Reviewer_Sg2F · 2025-08-02
>
> Thank authors for their response. Overall, it was helpful in improving my understanding of the paper. While I remain open to revising my evaluation, I would like to revisit a few concerns I raised earlier—particularly those the authors mentioned were unclear.
>
> One of my primary concerns is that the central idea of "reusing the KV cache" in aLoRA does not seem too novel. Specifically, I had asked whether it is not somewhat trivial to train aLoRA adapter per task while keeping the KV cache fixed. My understanding is that the Transformer + aLoRA setup should already have sufficient expressiveness to handle narrow-domain tasks, such as those discussed in the SFT experiment in Section 4.1.
>
> If I’m not mistaken, isn’t the comparison essentially between LoRA and aLoRA trained per task, with the main difference being that aLoRA reuses the backbone’s KV cache? In that case, the experiments mainly show that aLoRA is viable under this shared-cache setup? Hence, this is just verifying simply sharing the KV cache is possible.
>
> Second, I was trying to raise concerns regarding the invocation mechanism. From what I understand, the invocation logic is predefined, meaning that aLoRA parameters are loaded manually based on a fixed schedule or pattern. If that’s the case, the novelty in this aspect is pretty limiting. There doesn’t appear to be any dynamic selection of aLoRA parameters by the model itself.
>
> I would appreciate further clarification from the authors on the novelty aspect. Again, I do see value in the paper’s contributions—which is why I did not start with a clear rejection. My aim here is to better understand the scope and novelty of the work.
>
> ***Minor points***
>
> > **Response to "adaptation achievable?"**
>
> Thank you for the clarification. It is indeed encouraging to see that aLoRA is expressive enough to adapt to far-domain data distributions.

---

> ### Author Response · Authors · 2025-08-04
> **Response and clarifications**
>
> Thank you for replying, and being willing to revise your evaluation! Indeed, there seem to be some major disconnects. We trust that the below will clarify your understanding of aLoRA, the KV cache, and LLM inference in the context of our work.
>
> **One of my primary concerns is that the central idea of "reusing the KV cache" in aLoRA does not seem too novel...**
>
> These concerns seem to come from some kind of misunderstanding. We now realize we may have been slightly confusing in our use of the term "KV cache", since it may sound like a simple _storage location_ in memory --- while in fact we meant _literal re-use of the KV hidden states in the transformer neural network_. As such, our innovation is a novel _transformer adapter architecture_ yielding models not interchangeable with LoRAs at all. We will clarify this in revision.
>
> 1. **whether it is not somewhat trivial to train aLoRA adapter per task while keeping the KV cache fixed** This is not how adapters are trained, the KV cache is not part of training and is something only used during generation. During training, our architecture ensures that relevant KV values are interchangeable in the way we require. Setting this up required deep modification of large open source packages.
> 2. **isn’t the comparison essentially between LoRA and aLoRA trained per task, with the main difference being that aLoRA reuses the backbone’s KV cache? In that case, the experiments mainly show that aLoRA is viable under this shared-cache setup? Hence, this is just verifying simply sharing the KV cache is possible.** This is _not_ the comparison -- and it seems possible that you have misunderstood what we mean by KV reuse? The "KVs" are a set of saved hidden states in the Transformer architecture. "KV reuse" means that the aLoRA adapted transformer is being constrained so that important hidden states therein exactly match other hidden states in the separate non-adapted transformer. Firstly, this is diametrically opposed to how LoRA (and all prior LoRA literature) are designed. Second, it so happens that there is major _runtime_ reason to train the model to match these states, specifically because you no longer need to compute them but instead simply load them from prior base model operations. As we demonstrate, this creates an order of magnitude speedup with no loss of accuracy. We respectfully submit that speedups of this magnitude are not trivial at all, and are of interest to the community (particularly as they come from a novel adapter network architecture).
>
> **...the invocation logic is predefined, meaning that aLoRA parameters are loaded manually based on a fixed schedule or pattern...the novelty in this aspect is pretty limiting. There doesn’t appear to be any dynamic selection of aLoRA parameters by the model itself.**
>
> We don't view the "invocation" as anything exciting or "novel", it is simply a way of stating that the weights need to be activated _at some point_. We are agnostic to how that happens. We have pointed out many ways this can happen: (a) indeed manually (rare), (b) as part of an agentic flow created for a task (most common) -- see the vast literature on this, (c) via tool calling by an unstructured LLM. We emphasize that this last (c) type of invocation behavior is **already widely used in nearly all LLMs** at some level, with the most basic scheme being the chat template that nearly all models use.
>
> As the easiest example, consider the RAG setting which we discuss in the paper. First, note that RAG is an extremely common use case for LLM agents, getting major investment and attention in the literature --- not a narrow or limited use case. In this setting, for each user question, there is a very natural flow from querying the database or search engine, to preparing a response to the user. In our paper, we have proposed and empirically proven many aLoRA adapters directly relevant to this use case, which will directly improve all aspects of this task. For all of these, it is completely unambiguous where the "invocation" must happen, and is not an issue.
>
> For more complex agentic pipelines, adapters such as certainty, hallucination, or constraint checking, or correcting insufficient responses are universally relevant to LLM agents of all types. Again, it is usually not difficult to determine when these should be invoked (usually after some kind of base model generation), and simple rules can be created to invoke these when appropriate.
>
> Finally, we note the fully autonomous invocation is straightforward. Modern LLMs are well-trained to know when to invoke "tools" provided to them given a text description of what the tool does. Describing available aLoRA adapters to the model in this way would be trivial to do, and provide fully autonomous control of when to activate the aLoRA adapters.
>
> We hope the above has clarified your view of the paper! We would be glad to further clarify anything in the time that remains.

---

> > ### Author Response · Authors · 2025-08-06
> > **Checking in**
> >
> > Thanks again for the discussion! Given the window for discussion is growing short, we wanted to check in as to whether our last response clarified your questions. Please do let us know if anything remains unclear or if your perspective has changed. In particular, we remain very surprised by your assessment of our contribution as "trivial", and believe this is likely due to a misunderstanding we outline. If we are misunderstanding your concerns please let us know as well! We also encourage you to take a look at the other reviews and discussion happening there.

---

> ### Author Response · Authors · 2025-08-08
> **Checking in**
>
> Thanks again for the discussion! We are reaching out again since you indicated you were willing to adjust your evaluation, and the time is growing short. We believe that your review may be the tipping point for our work to be accepted. As such, please do let us know if there is anything remaining that keeps you from raising your score. We really appreciate your engagement with us and willingness to consider our work!

---

### Official Review · Reviewer_AjC2 · 2025-07-02

**Clarity:** 3
**Significance:** 2
**Originality:** 2
**Rating:** 3
**Confidence:** 4

**Summary:**

This paper proposes Activated LoRA (aLoRA), a method designed to significantly improve inference efficiency in scenarios where multiple LoRA adapters are dynamically invoked, such as in multi-turn agentic pipelines. The key contribution is a mechanism that allows selective application of adapter weights to only a portion of the input tokens, thereby enabling KV cache reuse and reducing redundant computation.

**Questions:**

1. What are the specific ranks used for LoRA and aLoRA in the supervised fine-tuning experiments? Since the rank directly impacts parameter count and memory usage, it would be useful to report this explicitly in the result tables.

2. How does aLoRA integrate with other widely-used LoRA extensions such as DoRA and QLoRA? Since these methods address different aspects (e.g., quantization, dynamic scaling), understanding how they interact with aLoRA would be valuable to practitioners.

**Ethical Concerns:**

["NO or VERY MINOR ethics concerns only"]

**Final Justification:**

The paper defines a scenario that may be practical for specific use cases and presents promising results within that scope. However, the methodological innovation appears modest, and it is debatable whether the proposed use cases are too constrained for some meaningful tasks.

**Limitations:**

The limitations and potential negative societal impact are not provided.

**Quality:**

2

**Strengths And Weaknesses:**

Strengths
1. The paper is clearly written and well-organized, making it easy to follow both the motivation and the technical details of the method.

2. The experimental results effectively demonstrate that aLoRA substantially reduces inference time compared to standard LoRA, especially in settings with long contexts and multiple concurrent adapters.

Weaknesses:
1. Limited Novelty. While the cache reuse mechanism is practically important, the idea is conceptually straightforward. The novelty lies more in the engineering and integration than in theoretical advancement.

2. The datasets used for SFT are relatively small. Demonstrating similar gains on larger-scale or more diverse instruction tuning datasets would make the findings more compelling.

---

> ### Author Rebuttal · Authors · 2025-07-30
>
> We thank the reviewer for their review and for acknowledging the effectiveness of our presentation and experiments. The reviewer's comments have helped us notice possible areas of improvement in our messaging, which we will address in the revision.
>
> > ”Limited Novelty. While the cache reuse mechanism is practically important, the idea is conceptually straightforward.”
>
> We strongly emphasize that "straightforwardness" should not be seen as a counterargument to "novelty." Indeed, the beauty of our innovation is that a straightforward, non-disruptive change can achieve such major practical runtime improvements. This is in fact an advantage that _magnifies_ the potential for impact in the field, as the "straightforwardness" of our approach is key to being able to integrate tightly with vLLM and other open source packages used by researchers.
>
> > ”The datasets used for SFT are relatively small.”
>
> This is not quite true. While some of the initial benchmark datasets are small, the datasets used for the intrinsics finetuning tasks are actually large. For instance, the certainty intrinsic uses hundreds of thousands of data points, which is an extremely large dataset in the context of LoRA finetuning.
>
> > ”What are the specific ranks used for LoRA and aLoRA in the supervised fine-tuning experiments? Since the rank directly impacts parameter count and memory usage, it would be useful to report this explicitly in the result tables.”
>
> We will add a table showing the ranks selected by grid search in the initial SFT benchmark experiment (the ranks used for the intrinsics SFT experiments are already shown). The ranks selected by grid search did not show any consistent patterns, in particular, there was no increase in the chosen rank for aLoRA overall. Even though the rank differences are not substantial, we decided to be conservative in our computational complexity experiments and chose, for the vLLM inference experiments we showed, to compare aLoRA with $r=32$ to LoRA with $r=8$. In other words, the very large speedups we show for aLoRA are with 4x the adapter parameters than the LoRA it is compared to. Our experiments strongly suggest that the adaptation rank is not contributing to the cost of generation. Indeed, it can be intuitively seen that the rank of the adapter will have almost no impact on computation and memory costs, as the adapters in all cases count for <1% of total model parameters.
>
> > ALoRA integration with QLoRA and DoRA
>
> Thanks for this observation! We will add a statement that QLoRA, being simply a quantization of the base model, is easily extended to the aLoRA setting (it is however beyond the scope of the present work to include experiments). The caveat is that for KV cache reuse to apply, the base model should also be running in the same quantization.
>
> Similarly, we can address DoRA. While in principle the same idea could be applied to DoRA, DoRA is not efficient to run as an adapter, and the standard packages recommend merging the learned weights to the base model weights before doing inference. As merging is not possible with aLoRA due to selective weight application, we don’t anticipate this to be a useful direction unless the inefficiencies of DoRA can be addressed or mitigated.
>
> > Limitations
>
> We will add a more explicit statement in the revision.
>
> **Summary**
>
> We hope that these clarifications and forthcoming additions will help address your concerns and merit a reconsideration of your evaluation. If concerns remain, please let us know during the discussion period!

---

> ### Author Response · Authors · 2025-08-05
> **Do you have further questions?**
>
> Thanks again for reviewing our work. Please let us know if our rebuttal addresses your concerns, or if you have further concerns we can address.
>
> Best,
> the authors

---

> ### Comment · Reviewer_AjC2 · 2025-08-07
>
> Thank you to the authors for their response. After reviewing your reply and considering comments from other reviewers, my concern remains that the methodological innovation appears modest, thus I will keep my score unchanged. In my view, the primary contribution of the paper lies in defining a constrained scenario that may be practical for specific use cases; however, the methodology applied to this scenario is relatively straightforward.
>
> In your response to other reviewers, you state that "aLoRA matches (Hindi) and exceeds (Bengali) LoRA." However, Figure 4 shows a substantial performance drop for aLoRA compared to LoRA in the Hindi Sentence Perturbation task on Llama 3.2 3B and 8B. While this gap might be attributed to dataset size, the results make it difficult to conclude that aLoRA truly matches LoRA for Hindi or exceeds it for Bengali. On the contrary, we should be cautious about assuming that LLMs will "magically" adapt to unseen languages/tasks.

---

> > ### Author Response · Authors · 2025-08-07
> > **Clarification**
> >
> > We point out that the Llama 3.2 is not relevant to this question, as both Llama models were pretrained on these languages. As we noted in the paper, we don't attempt to draw major conclusions from any individual experiment here, as the datasets are small. We certainly do not assume any "magical" adaptation, but merely contend that aLoRA is capable of handling instances where the test language is not familiar to the base model. As the reviewer raised this question, implying that this adaptation was impossible, _any_ success story in such a regime proves that this is not an automatic failure case for aLoRA.
> >
> > We also strongly push back against a "modest innovation" evaluation. The 10X to 30X speed improvements speak for themselves. This is hardly an insignificant improvement, and it results from a novel architectural innovation that was inspired by a deep understanding of the models involved.

---

### Official Review · Reviewer_jjBF · 2025-07-03

**Clarity:** 3
**Significance:** 3
**Originality:** 3
**Rating:** 4
**Confidence:** 3

**Summary:**

This paper introduces Activated LoRA (aLoRA), a novel variant of Low-Rank Adaptation (LoRA) designed for efficient multi-turn usage of large language models (LLMs).  LoRA fine-tuning updated the model weight with a low rank matrix. However, when switching between multiple LoRA adapters in a conversation or agent pipeline, one must currently recompute the model’s key-value (KV) cache for the entire context each time a new LoRA is applied during inference, which is inefficient for long conversations. To address this, the authors propose aLoRA, an adapter architecture that activates the LoRA weights only for tokens after a designated “intrinsic” invocation point. In effect, the base model processes the earlier context normally (caching its representations), and the LoRA adapter is turned on mid-stream to handle a specific sub-task. This design allows reusing the base model’s existing KV cache for all tokens before the adapter’s activation, avoiding any recomputation of the earlier context. Experiments and simple analysis are provided to corroborate the idea.

**Questions:**

How do you compare this aLoRA method with prompt tuning, which also seems to support KV cache reuse. In the current agent settings, a huge amount of the pipeline/workflow construction is done by prompt designing (and possibly prompt tuning). Can you think about some cases where it is hard to use prompt tuning (e.g., just append a specific prompt after the content to use a specific feature). Do you think your method can support smaller models than prompt tuning (which may mainly focus on large models)?

In a complex pipeline, one might invoke several different intrinsics in sequence or even in parallel (as evaluators). Can you elaborate on how aLoRA handles back-to-back or concurrent intrinsic calls? For example, if intrinsic A generates some text that is then used as input to intrinsic B (or back to the base model), do we simply treat that output as new input (incurring a base model pass for B’s input since base didn’t see those tokens)? In Figure 2 and your description, it seems base-model-to-intrinsic reuse is excellent, but intrinsic-to-base reuse is not supported (since aLoRA’s own generated tokens aren’t in the base’s cache).

**Ethical Concerns:**

["NO or VERY MINOR ethics concerns only"]

**Final Justification:**

The authors wrote long responses, and lots of them address my concern, and make me understand better about their contribution. However, I don't feel like it is a score 5 paper at this point, but I would also like to keep the positive score. Thus, I will maintain my score unchanged, but increase my confidence slightly.

**Limitations:**

I do not see a limitation section in the main paper. Maybe it is possible to include sth related to the model size for the experiments.

**Quality:**

3

**Strengths And Weaknesses:**

Strengths:

The paper tackles the real-world inefficiency of using multiple LoRA fine-tunes in one session. This dynamic adapter activation is a novel solution that directly targets the latency overhead in multi-skill LLM applications (e.g. agent systems, tool-augmented chats).

Activated LoRA is, to the best of my knowledge (I am not an expert in this specific field about KV cache optimization and inference speedup), a new extension to the LoRA paradigm. Instead of static adapters that always apply to the whole input, aLoRA’s “activate on trigger” approach is original. It essentially treats adapters like callable functions at inference time, which is an innovative perspective on model modularity.

The paper’s experiments are comprehensive. It evaluates aLoRA on two fronts: (1) Inference speed in a realistic multi-step pipeline, and (2) Task performance on both standard benchmarks and new intrinsic tasks.

Weakness:

The solution is somewhat tailored to scenarios where distinct task segments can be isolated in the prompt (the “intrinsic” is clearly delineated from the main conversation). This covers many use cases (the paper focuses on QA/RAG pipelines, agentic evaluators, etc.), but there are cases where a task might require full-context adaptation. For instance, if one wanted to use a LoRA to improve how the model interprets the entire user query (say for better understanding of domain-specific language), aLoRA’s approach of not adapting earlier tokens could be suboptimal.

Although the overall trend shows parity between aLoRA and LoRA, some individual results exhibited noticeable difference. The authors claim that certain tasks had noisy outcomes due to small data sizes or non-exhaustive hyperparameter search. While they found no systematic disadvantage to aLoRA (median difference ~0%), these variations raise questions. It suggests that aLoRA may occasionally require more careful tuning (or higher rank) to match LoRA on certain tasks.

The experiments are conducted on models up to ~8–14 billion parameters (with many on 1B–7B). It is understandable due to computational constraints, but it means we do not see results on very large-scale LLMs. It is unclear if the method works well on larger scale. (Note: this is a minor issue, and should not be viewed as a concern for rejection. Even if there is no exp for larger model, the small model exps are already interesting)

---

> ### Author Rebuttal · Authors · 2025-07-30
>
> We thank the reviewer for their insightful review, appreciation of our proposed setting, and positive evaluation of the method and its originality. We respond to questions below. If anything remains unclear, let us know!
>
> > “The solution is somewhat tailored to scenarios where distinct task segments can be isolated in the prompt (the “intrinsic” is clearly delineated from the main conversation). This covers many use cases (the paper focuses on QA/RAG pipelines, agentic evaluators, etc.), but there are cases where a task might require full-context adaptation. For instance, if one wanted to use a LoRA to improve how the model interprets the entire user query (say for better understanding of domain-specific language), aLoRA’s approach of not adapting earlier tokens could be suboptimal.”
>
> Regarding interpreting an entire user query in domain-specific language – note that our benchmark experiments already involve this scenario, specifically the Hindi and Bengali based benchmarks, for which Mistral had little to no training (see relevant online materials). aLoRA matches (Hindi) and exceeds (Bengali) LoRA. We therefore do not see “unfamiliar prompt” to be a particularly difficult case for aLoRA, though more difficult regimes could exist. We also include an intrinsic (e.g. Query Rewrite) where careful understanding of the entire prior context is important to create a self-contained rewritten query. A hypothesis is that aLoRA is exploiting the fact that LLMs have evolved to provide powerful, if not task specific, representations of contexts that can be tweaked to new unseen tasks at runtime.
>
> That said, we agree that aLoRA is likely not universally superior to LoRA in all scenarios, indeed in inference pipelines where one adapter is used exclusively and the base model never invoked, LoRA will naturally nearly always be preferred, since there is no need for cross-model cache reuse. That said, we believe we have identified a wide range of highly interesting applications where the approach will make a very meaningful impact in runtime efficiency, especially as agentic systems move towards becoming ubiquitous.
>
> > ”some individual results exhibited noticeable difference [between aLoRA and LoRA]”
>
> Note that this was only observed in the case of the small benchmarks, where (a) some of the datasets are very small and (b) we had to limit ourselves to pre-defined hyperparameter grid search. Note that for all “real-world” tuning applications where we were able to invest more in tuning the model, we do not see these differences. That said, we have already called out in the paper that aLoRA does often require higher rank and slightly more careful tuning, but we have found these downsides to be surmountable without too much effort. In the revision, we can revisit the handful of failure cases here and do a denser hyperparameter grid search to present a fuller picture.
>
> > ”It is understandable due to computational constraints, but it means we do not see results on very large-scale LLMs…this is a minor issue”
>
> Thanks for raising this point (and observing that the answer is indeed our computational constraints). We tested the largest models that we could fit on our H100 GPU. We attempted multi-GPU inference with larger models, but at time of submission, vLLM v1 with both LoRA and aLoRA was failing on our system (note that vLLM v1 is in very active development). That said, we will revisit this and hope to get larger multi-GPU vLLM experiments in time for the revision. Based on everything we have seen, we have very strong reasons to believe all observed trends will continue, as prefill costs remain roughly similar in a relative sense between small and large models.
>
> > Comparison with prompt tuning
>
> There are several prompt tuning methods in the literature. As we mention in the paper, while the most performant prompt tuning methods are generally _not_ supportive of cache reuse since they tune the prefix, tuning the suffix will preserve KV reuse in a meaningful way and may be appropriate for certain intrinsics. We note, however, that it is widely established in the literature (e.g. He, Junxian, et al. "Towards a unified view of parameter-efficient transfer learning." arXiv preprint arXiv:2110.04366 (2021)) that prompt tuning methods underperform LoRA-based methods, often significantly. As a result, the innovation of aLoRA remains important to achieve the best performance for these specialized tasks in general. In the revision, we will more clearly make this point and re-emphasize that aLoRA is not the only solution that enables KV cache reuse, but rather a way to unify the accuracy advantages of weight tuning with the runtime advantages of late prompting (whether fixed or optimized). We will expand our discussion of this point in the revision.
>
> > Intrinsics in sequence and prefill
>
> You are correct that _generated_ aLoRA tokens will have KV cache not usable by the base model or other adapters. We believe this fairly unavoidable, and is a fairly minimal downside since the aLoRA generated tokens can be simply prefilled into the base model cache after generation, if needed. We say this is a minor downside because (a) for many intrinsics, the output of the aLoRA is not needed for subsequent base model calls, (b) the generation of the aLoRA will usually be short, or at least much shorter than say an input document or base model generation, (c) prefilling short strings is very fast, much faster than generating that same string. As a result, the overall cost of generating with the aLoRA and then prefilling the result into the base model is a small extra cost proportionally to the cost of the aLoRA generation (and is often not needed).
>
> Finally, recall that _prefilled_ tokens done by an aLoRA adapter _are_ reusable by the base model and any other aLoRA. As a result, the main gains of reuse are in fact achieved in the reverse direction (intrinsic-to-base and intrinsic-to-intrinsic).
>
> > Limitations
>
> The primary limitation that we called out in the paper is that aLoRA often needs higher rank $r$ than LoRA. As we noted in the paper, while this may expand (already small) training costs, this has no measurable downside in terms of inference costs or model accuracy. We will add this to the conclusion.
>
> **Summary**
>
> Thanks again for the interesting suggestions, we hope that these clarifications and forthcoming additions will help address your concerns and merit an "accept" rating.

---

> ### Comment · Reviewer_jjBF · 2025-08-03
>
> Thank you for the rebuttal. I will keep my score (as my previous score is already positive).

---

### Note · Authors · 2025-08-11

**Final Remarks to AC**

We appreciate your efforts in guiding this process and want to make your decision as straightforward as possible. Below is a clear status update on each reviewer (as we see it) and our closing case.

**mmG2 (5 – accept):** Strongly positive on real-world significance, originality, and empirical validation. Only concern was applicability to complex prompts; we demonstrated these naturally fit our framework. Score unchanged, already positive.

**jjBF (4 – weak accept):** Positive throughout, highlighting novelty, comprehensive experiments, and relevance. Minor scope concerns were fully addressed in rebuttal; score remained positive.

**AjC2 (last seen 3):** Concerned about “modest innovation” and a misreading of our Hindi/Bengali results. We clarified that the large 10–30x speedups come from a *novel architectural change* enabling **KV reuse**—a capability fundamentally at odds with prior LoRA designs and requiring deep integration with training and inference pipelines. Score unchanged.

**Sg2F (last seen 3):** Initially viewed contribution as “trivial” likely due to a misunderstanding of the method, possibly thinking KV reuse was simply sharing a memory location. We clarified that our method enforces exact hidden-state alignment between adapted and base models mid-sequence—a wholly novel concept that was nontrivial to design and implement and yielded major improvements. Reviewer stated they were open to raising their score but did not follow up.

**Case for Acceptance:**
- **Clear Novelty:** This is the first adapter architecture to allow *instant mid-sequence activation with true KV state reuse*, enabling adapters to interoperate without re-prefilling history—something LoRA and its variants cannot do.
- **Significance:** Solves a key bottleneck for modular multi-adapter, multi-turn agentic systems—an increasingly important application domain.
- **Impact:** Up to 10–30× real-world speedups without accuracy loss, validated across multiple realistic pipelines and benchmarks.
- **Broad Applicability:** Method is not tied to “evaluation” or "rewrite" use cases; it applies wherever modular adapters are beneficial.

Two reviewers are already positive; the other two objections stem from misunderstandings we directly resolved in discussion. Given the strong technical contribution, large empirical gains, and clear community relevance, we believe the paper meets the NeurIPS bar for acceptance.

Thanks for your consideration!

---

### Decision · Program_Chairs · 2025-09-17

**Decision:**

Accept (poster)

**Comment:**

This paper introduces  Activated LoRA (aLoRA), effectively (to my understanding) allowing for "hot swapping" LoRA adapters mid processing, allowing these adapters to reuse some of the base model's work.

During the review phase, there was generally disagreement about this paper, with two reviewers highlighting empirical validation, significance, and good novelty and two reviewers suggesting "low novelty." I am recommending obvious acceptance here. I generally think this is a case where the negative reviews just don't really make any compelling arguments for rejection at all. To me, "low novelty" needs to mean "extremely similar / increlemental relative to existing prior work," to be grounds for rejection by itself, not "conceptually straightforward" as Reviewer AjC2 explicitly said their concern was.

In general, conceptually straightforward methods in machine learning that work are strictly better than the alternative, and I don't think conceptual complexity is the thing we should be hill climbing for at NeurIPS. Since there weren't really any non-superficial concerns raised by the rejecting reviewers, I'm aligning with the other two.